# Prenatal Exposure to Preeclampsia and Long-Term Ophthalmic Morbidity of the Offspring

**DOI:** 10.3390/jcm9051271

**Published:** 2020-04-28

**Authors:** Eliel Kedar Sade, Tamar Wainstock, Erez Tsumi, Eyal Sheiner

**Affiliations:** 1Faculty of Health Sciences, Joyce and Irving Goldman Medical School, Ben-Gurion University of the Negev, Beer-Sheva 8410501, Israel; 2Department of Public Health, Faculty of Health Sciences, Ben-Gurion University of the Negev, Beer-Sheva 8410501, Israel; 3Department of Ophthalmology, Faculty of Health Sciences, Soroka University Medical Center, Ben-Gurion University of the Negev, Beer-Sheva 8410501, Israel; 4Department of Obstetrics and Gynecology, Soroka University Medical Center, Ben-Gurion University of the Negev, Beer-Sheva 8410501, Israel

**Keywords:** preeclampsia, long-term morbidity, ophthalmic morbidity, pregnancy

## Abstract

The aim of this population-based study was to evaluate whether prenatal exposure to preeclampsia poses a risk for long-term ophthalmic morbidity. A population-based cohort analysis compared the risk of long-term ophthalmic morbidity among children who were prenatally exposed to preeclampsia and those who were not. The study population was composed of children who were born between the years 1991 and 2014 at a single tertiary medical center. Total ophthalmic hospitalization and time-to-event were both evaluated. A Kaplan–Meier survival curve was conducted to compare cumulative ophthalmic hospitalization incidence based on the severity of preeclampsia. Confounders were controlled using a Cox regression model. A total of 242,342 deliveries met the inclusion criteria, of which 7279 (3%) were diagnosed with mild preeclampsia and 2222 (0.92%) with severe preeclampsia or eclampsia. A significant association was found between severe preeclampsia or eclampsia and the risk of long-term vascular-associated ophthalmic morbidity in the offspring (no preeclampsia 0.3%, mild preeclampsia 0.2% and severe preeclampsia or eclampsia 0.5%, *p* = 0.008). This association persisted after controlling for maternal age and ethnicity (adjusted hazard ratio (HR) 1.861, 95% CI 1.051–3.295). In conclusion, within our population, prenatal exposure to severe preeclampsia or eclampsia was found to be a risk factor for long-term vascular-associated ophthalmic morbidity in the offspring.

## 1. Introduction

Preeclampsia is a disorder that complicates up to 7% of all pregnancies and is a leading cause of morbidity and mortality of both mother and fetus [1,2]. Throughout the years, many hypotheses have been proposed to explain the pathophysiological mechanism underlying preeclampsia. One leading hypothesis suggests that an impairment in the placentation process leads to a hypoxic placental state, which triggers a systemic endothelial dysfunction and consequently affects various body systems [3,4].

An alternative hypostasis endorses that the manifestation of preeclampsia is predisposed by impaired cardiovascular conditions in the mother. As the pregnancy becomes increasingly demanding, the failure of the cardiovascular system to properly adapt subsequently leads to a secondary placentation dysfunction, which in turn leads to the development of preeclampsia and systemic endothelial dysfunction [5,6].

Nevertheless, it is widely agreed that the abnormal development of the placenta stands at the core of the pathophysiology of preeclampsia. Impaired placentation leads to placental hypoxia, which subsequently increases the expression of hypoxia-inducible factor-1 (HIF-1), a transcription factor that plays a central role in the induction of systemic endothelial dysfunction in both the mother and the fetus [4,7,8]. This pathological endothelial state consequently affects distant tissues and body systems in both the mother and the fetus, including cardiovascular, renal, hepatic, neurological and visual systems, as up to 25% of mothers with severe preeclampsia suffer from visual symptoms [3,9,10,11,12,13].

Studies have found that the repercussions of the systemic endothelial dysfunction of preeclampsia continue to have an affect years after delivery [14]. Women with a history of preeclampsia were predisposed to several long-term maternal vascular morbidities, such as cardiovascular, renal and chronic hypertension [10,15,16]. A recent study has also found a significant risk for long-term vascular ophthalmic morbidities, such as diabetic retinopathy and retinal detachment [17]. This knowledge has recently led the International Federation of Gynecology and Obstetrics (FIGO) to recommend the follow-up of all women with preeclampsia 6–12 weeks after birth, and periodically thereafter, with screenings for hypertensive disorders and cardiovascular risk factors [18].

As for the offspring, the hypoxic condition of the placenta causes great stress to the intrauterine environment and poses a significant risk for immediate neonatal morbidity and mortality [19,20]. The definitive treatment of preeclampsia is the delivery of the fetus, therefore it is considered to be a leading cause of preterm delivery that results in low birth weight and short-term visual morbidity, such as retinopathy of prematurity (ROP) [19,21].

Several hypotheses suggest that preeclamptic intrauterine stress triggers an adaptive epigenetic programming reaction in the fetus that permanently alters gene expression and increases the susceptibility of the child to vascular diseases later in life [20,22,23,24]. Offspring who were prenatally exposed to preeclampsia have an increased risk for long-term vascular morbidities, such as chronic hypertension, stroke and cardiovascular diseases [10,22,23,25].

The short-term ophthalmic complications of preeclampsia for both mother [11,13] and child [19] have been previously described in the literature, focusing mostly on retinopathy of prematurity of the offspring. While recent studies addressed the long-term ophthalmic morbidity in women with a history of preeclampsia [17], knowledge of the long-term ophthalmic morbidities in the offspring is yet to be established. This gap of knowledge is mainly due to a lack of published data regarding the long-term ophthalmic consequences in children up to the age of 18, who were born to mothers with preeclampsia. Being a multisystem vascular disorder, we proposed a hypothesis that the visual system of the offspring (regardless of gestational age) is also affected by the long-term outcomes of preeclampsia. The purpose of this population-based study was to evaluate whether preeclampsia poses a risk for long-term ophthalmic morbidity in the offspring.

## 2. Experimental Section

This retrospective population-based cohort study was based on non-selective population data, as it was conducted at Soroka University Medical Center (SUMC), a sole tertiary medical center that serves the population of the Negev region in Israel, a region that spans the entire southern half of the State of Israel. The study protocol was approved by the institutional review board (SUMC IRB Committee), in accordance with the Helsinki declaration and informed consent was exempt. All singleton deliveries between the years 1991 and 2014 were included in the study population. Exclusion criteria were multiple gestations, perinatal mortality and offspring with congenital and chromosomal abnormalities.

The study compared offspring who were prenatally exposed to preeclampsia and those who were not. The exposure to preeclampsia was divided into two categories based on the severity: mild preeclampsia (International Classification of Disease, 9th revision (ICD-9) codes 64242, 64241) versus severe preeclampsia (ICD-9 codes 64252, 64251) and eclampsia (ICD-9 codes 64261, 64262). Due to the small number of cases of eclampsia, (*n* = 75) it was included in the severe preeclampsia category. The severity of preeclampsia was classified according to the guidelines of the Working Group of the National High Blood Pressure Education Program [26].

The main outcome was subsequent offspring ophthalmic morbidity. Thus, we conducted a retrospective follow-up of offspring up to the age of 18 years who were hospitalized due to ophthalmic conditions. The follow-up time started immediately after their release from hospital and was defined as time to an event (first hospitalization due to ophthalmic condition), or until censored in the case of mortality or at the age of 18 years. Ophthalmic hospitalizations were defined in accordance with the International Classification of Disease, 9th revision (ICD-9), and included conditions such as visual disturbance, ophthalmic infection, inflammation and retinopathy of prematurity (ROP) documented at SUMC (see Table A1 and Appendix A). The data collected were the result of merging and crosslinking between two different databases. The first set of data was based on the collection of computerized perinatal data that had previously been recorded at the delivery stage by the obstetrician. The second database was based on the collection of pediatric hospitalization records at SUMC. This computerized database included both demographic information and all medical diagnoses that were made during hospitalizations at SUMC, and were classified under the ICD-9 codes. Following delivery, all offspring were issued with a national security ID assigned by the state. These ID numbers were also registered under the mother’s identification card. Entering both mother and offspring ID numbers allowed us to track and link the relationship between mother and offspring in our database.

### Statistical Analysis

We performed a statistical analysis using the SPSS package, version 23 (IBM/SPSS, Chicago, IL, USA). Chi-square and analysis of variance (ANOVA) tests were used to identify statistical differences for general association. Continuous variables with normal distribution (maternal age and gestational age) were presented in years as mean ± standard deviation (SD) and were evaluated using one-way ANOVA test. Categorical variables with non-normal distribution (parity, preterm delivery, low birth weight, ethnicity and Apgar score) were presented in percentages and were evaluated using a Pearson chi square test. The categories were defined as accepted in the prenatal literature. Parity was categorized into three categories (1, 2–4, 5+) in order to reflect both nulliparity and grand multiparity, as it is known to have a clinical effect [27]. Apgar was categorized into two categories: greater than or equal to seven points and less than seven points, in accordance with its clinical significance [28]. Birth weight was categorized into two categories: greater than or equal to 2500 g, and less than 2500 g, in accordance with its clinical significance. Since preterm delivery is a major outcome in preeclampsia, we entered gestational age not only as a continued variable, but also as a categorical variable (i.e., preterm before 37 or before 34 week of gestation).

The date of the first hospitalization was used to specifically calculate time-to-event in the survival analysis and to compare the differences in incidence between the study groups. Both Kaplan–Meier curves and Cox proportional hazard regression models were used in our study to detect the univariate and multivariable analyses. The cumulative ophthalmic hospitalization incidence over time was analyzed using a univariate statistical analysis performed by the Kaplan–Meier survival curve estimation. A multivariable Cox proportional hazard regression model was fit to estimate the associations between preeclampsia and long-term ophthalmic hospitalizations, while controlling for statistically significant and clinically important confounders. Maternal age and ethnicity were defined as confounders and were included in the final presented models. Maternal age is a clinically important risk factor for adverse perinatal outcome [29]. Additionally, ethnicity is a strong marker that highly correlates with socioeconomic status. The ethnic diversity of the two socioeconomic societies inhabiting the southern part of Israel, Jewish and Bedouins, represents a wider difference in the utilization of healthcare services, degree of income and access to resources [30,31]. From the models, the hazard ratio (HR) and 95% confidence interval (CI) as the effect measure were derived. The proportionality of the variables in the models was evaluated by visualization of the Kaplan–Meier graphs of each of the variables. A *p*-value of < 0.05 was considered statistically significant.

## 3. Results

A total of 242,342 deliveries were documented during the study period (Figure 1). After excluding perinatal morbidities (*n* = 1340), congenital and chromosomal abnormalities (*n* = 6150) and multifetal pregnancies (twins or triplets) (*n* = 11,454), 7279 (3%) cases were diagnosed with mild preeclampsia and 2222 (0.92%) with severe preeclampsia or eclampsia. The median follow-up time was 10.4 years in the no preeclampsia group, and 11.7 and 11.4 years in the severe preeclampsia or eclampsia group, respectively. During the follow-up time, 52,000 cases reached the age of 18 years and thus were defined as censored. The characteristics of the study population, based on the severity of preeclampsia, are shown in Table 1. Women diagnosed with preeclampsia were significantly older when compared to the no preeclampsia (PE) group (no PE 28.13 ± 5.79 vs. mild PE 28.66 ± 6.31 vs. severe PE or eclampsia 29.00 ± 6.94, *p* < 0.001). The rate of nulliparity was significantly higher among the mild and severe preeclampsia groups (no PE 22.9% vs. Mild PE 39.2% vs. Severe PE or eclampsia 43.4%, *p* < 0.001). Higher rates of preterm delivery (no PE 6.1% vs. mild PE 9.6% vs. severe PE or eclampsia 43.4%, *p* < 0.001) and low birth weight (no PE 5.9% vs. mild PE 10.6% vs. severe PE or eclampsia 46.2%, *p* < 0.001) were noted in offspring in the preeclampsia groups.

Table 2 presents the incidence of long-term ophthalmic morbidities between the study groups. Significant differences were noted between offspring prenatally exposed to preeclampsia (no preeclampsia, mild and severe preeclampsia) and long-term vascular-associated ophthalmic morbidities (0.3%, vs. 0.2% vs. 0.5% respectively, *p*-value = 0.008).

Figure 2 demonstrates the Kaplan–Meier survival curves for the cumulative incidence of long-term total ophthalmic hospitalization among the three study groups (no preeclampsia, mild preeclampsia and severe preeclampsia or eclampsia). The difference between the study groups was not statistically significant (Log-rank test, *p* = 0.118).

Figure 3 presents individual Kaplan–Meier curves to illustrate the cumulative incidence among the study groups for the following individual outcomes: vascular and other (Figure 3A), visual disturbance (Figure 3B) and infection and inflammation (Figure 3C). Prenatal exposure to severe preeclampsia was noted as a significant risk factor for vascular-associated ophthalmic hospitalization (Figure 3A, Log-rank test, *p* = 0.007).

Table 3 presents Cox regression models estimation that were used to examine an independent association between long-term ophthalmic morbidity of the offspring based on the exposure status. After controlling for maternal age and ethnicity, which were defined as confounders, prenatal exposure to severe preeclampsia or eclampsia was found to be a significant risk factor for vascular-associated ophthalmic morbidities in the offspring later in life (adjusted HR 1.861, 95% CI 1.051–3.295).

## 4. Discussion

Preeclampsia is a pregnancy-specific syndrome with multisystem involvement that increases the risk for vascular diseases in both a mother and her offspring [23]. The main goal of our study was to further investigate whether preeclampsia poses a risk for long-term ophthalmic morbidity in the offspring. The major findings of our study indicate that, in our population, severe preeclampsia or eclampsia were found to be a significant risk factor for long-term vascular-associated ophthalmic morbidity in offspring. Our findings are supported by several studies which have previously described the relationship between the severity of preeclampsia and long-term vascular morbidities (i.e., cardiovascular, renal) in relation to the mother [16] and the fetus [22]. Nevertheless, in our population, mild preeclampsia was not found to be a risk factor for long-term vascular-associated ophthalmic morbidity in the offspring. Additional research is required in order to further investigate these findings.

Many hypotheses have been proposed to explain the pathophysiological mechanisms by which preeclampsia increases the susceptibility for long-term morbidity in the offspring. Even so, our current understanding regarding these mechanisms is still limited. In concurrence with the Fetal Origins Hypothesis (also known as “Barker’s Hypothesis”), one hypothesis suggests that the hypoxic intrauterine environment triggers adaptive reactions in the fetus that might play a key role in the development of the long-term consequences of preeclampsia. The hypothesis explains that the stressed intrauterine environment leads to profound epigenetic programming that results in altered regulatory gene expression and impaired transcriptional activity in the fetus. This effect seems to extend into adulthood and permanently increases the susceptibility of the child to vascular morbidity later in life. The hypothesis is supported by various recent studies that have found preeclampsia to be an independent risk factor for long-term vascular morbidities, such as cardiovascular morbidity and a higher risk of hypertension [10,22,23,24,32,33].

The main strength of our population-based study is that our hospital provided both ophthalmological and obstetrical services to the entire population of southern Israel, as it serves as the sole hospital of the entire region. Additionally, the registration methodology in our databases allowed us to easily track and link the relationship between mothers and their offspring. Both strengths provided us with longitudinal data from a large cohort and allowed us to compare between different groups, based on the severity of the preeclampsia.

Nevertheless, several limitations of the study should be noted. Firstly, due to the lack of data on immigration in our study, patients who withdrew from the study before the end of the follow-up period were not counted in the study. In any case, there is no reason to suspect differential rates of outcome ascertainment in the study groups, as we can reasonably assume that such withdrawal was relatively equal in all study groups. Secondly, due to the retrospective nature of the study, only ophthalmic morbidities that resulted in hospitalization were recorded in our database. Therefore, it is reasonable to assume that most ophthalmic morbidities were treated in an outpatient setting. Generalizability can be addressed as one of the potential limitations of the study, as only ophthalmic morbidities that resulted in hospitalization were recorded in our database. The external validity of our findings may be limited to the severe cases of ophthalmic morbidities, those requiring hospitalization, and not to all ophthalmic morbidities, which are generally treated in an outpatient setting. However, due to the wide geographical area served by Soroka University Medical Center (SUMC), and the fact that all health services are fully covered by law for all citizens, there is no reason to assume that offspring requiring hospitalization would not show up to be treated at SUMC. Importantly, this is a large population-based study, so our finding can be useful and representative of the severe morbidities in other populations.

In addition, the rate of pediatric ophthalmic morbidity was significantly low (1.0%, *n* = 2330), as only severe cases requiring hospitalization were entered into our study. Therefore, despite our large sample size, our findings were limited to relatively small numbers of cases of ophthalmic hospitalization that were available to analyze, with limited statistical power to detect the effects on specific ophthalmic morbidities. Nevertheless, we were able to find significant difference between mild and severe preeclampsia and vascular-associated ophthalmic morbidity.

## 5. Conclusions

In conclusion, the present results of our study suggest that there is a significant association between prenatal exposure to severe preeclampsia or eclampsia and long-term vascular-associated ophthalmic morbidity in the offspring. Further studies are needed to determine the association between mild preeclampsia and the development of long-term vascular-associated ophthalmic morbidity in the offspring.

## Figures and Tables

**Figure 1 jcm-09-01271-f001:**
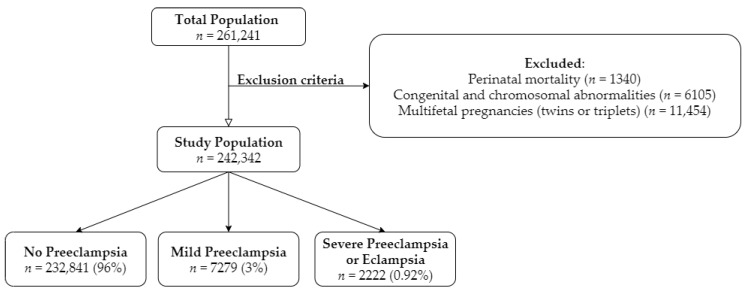
Presents the flowchart for study selection.

**Figure 2 jcm-09-01271-f002:**
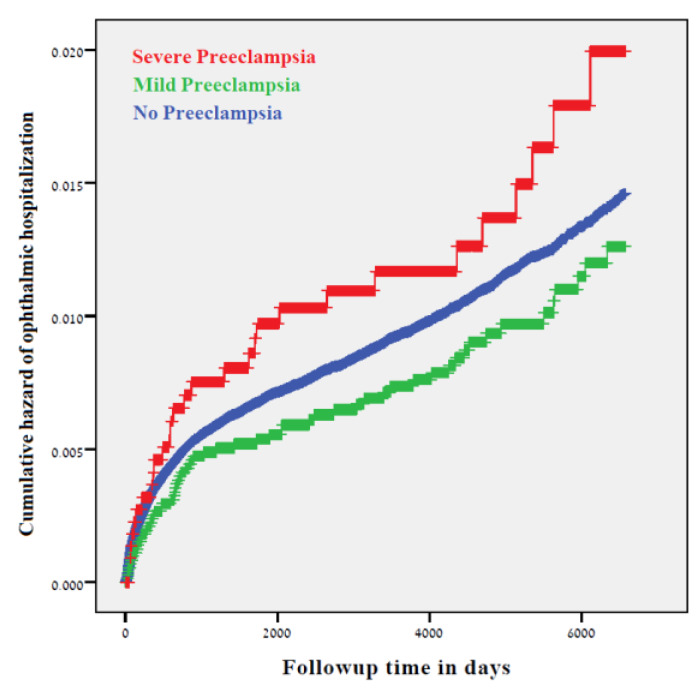
A Kaplan–Meier curve demonstrating the cumulative incidence of long-term ophthalmic morbidity in offspring exposed to preeclampsia (Log-rank test, *p* = 0.118).

**Figure 3 jcm-09-01271-f003:**
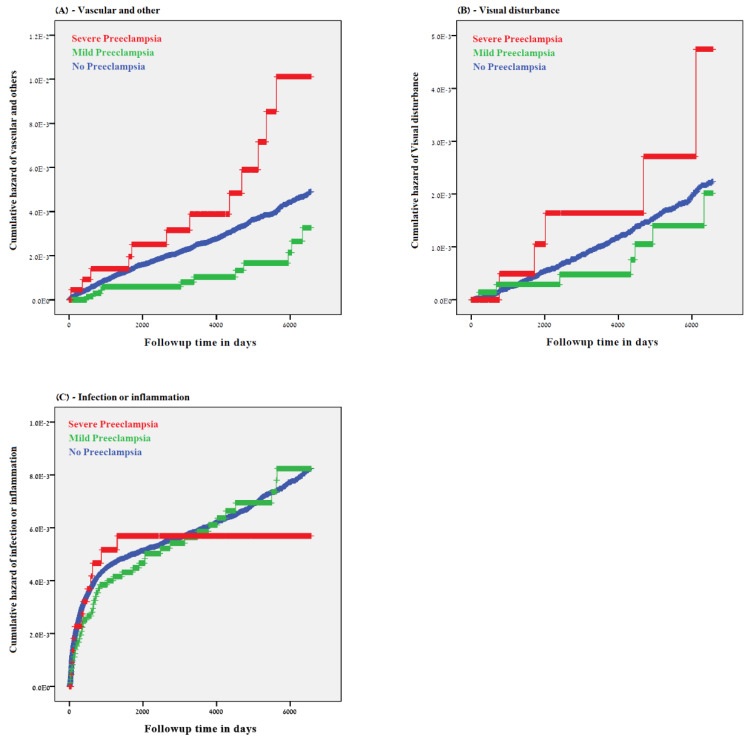
Kaplan–Meier curves demonstrating the cumulative incidence of long-term ophthalmic morbidity in offspring exposed to preeclampsia for each subcategory outcome: vascular and other (**A**), Log-rank test, *p* = 0.007, visual disturbance (**B**), Log-rank test, *p* = 0.303, and infection and inflammation (**C**), Log-rand test, *p* = 0.881.

**Table 1 jcm-09-01271-t001:** Clinical characteristics of the study population by exposure to maternal preeclampsia.

Maternal and Newborn Characteristic	No Preeclampsia *n* = 232,841	Mild Preeclampsia *n* = 7279	Severe Preeclampsia *n* = 2222	*p*-Value ^1^
Maternal age (years, mean ± SD)	28.13 ± 5.79	28.66 ± 6.31	29.00 ± 6.94	<0.001
Ethnicity (%)				
Jewish	109,751 (47.1%)	4136 (56.8%)	978 (44.0%)	<0.001
Bedouins	123,090 (52.9%)	3143 (43.2%)	1244 (56.0%)	
Parity (%)				
1	53,348 (22.9%)	2854 (39.2%)	964 (43.4%)	<0.001
2–4	120,382 (51.7%)	2903 (39.9%)	657 (29.6%)	
5+	59,060 (25.4%)	1502 (20.9%)	601 (27%)	
Gestational age (weeks, mean ± SD)	39.17 ± 1.82	38.71 ± 1.8	36.66 ± 3.01	<0.001
Preterm (<37)	14,150 (6.1%)	701 (9.6%)	965 (43.4%)	<0.001
Preterm (<34)	2212 (1.0%)	55 (0.8%)	328 (14.8%)	<0.001
Low birth weight (<2500 g)	13,668 (5.9%)	770 (10.6%)	1026 (46.2%)	<0.001
Apgar scores <7 at 1 min (%)	11,035 (4.7%)	349 (4.8%)	352 (15.8%)	<0.001

^1^ Data on maternal age and gestational age were evaluated using one-way analysis of variance (ANOVA) test and are presented in years as mean ± standard deviation (SD). All other data were evaluated using a Pearson chi square test and are presented in numbers and percentages.

**Table 2 jcm-09-01271-t002:** Long-term ophthalmic morbidity of offspring prenatally exposed to preeclampsia.

Ophthalmic Morbidity	No Preeclampsia *n* = 232,841	Mild Preeclampsia *n* = 7279	Severe Preeclampsia *n* = 2222	*p*-Value ^1^
Vascular and others	645 (0.3%)	11 (0.2%)	12 (0.5%)	0.008
Visual disturbance	271 (0.1%)	7 (0.1%)	5 (0.2%)	0.286
Infection\inflammation	1411 (0.6%)	45 (0.6%)	12 (0.5%)	0.915
Retinopathy of prematurity (ROP)	3 (<0.01%)	0 (-)	1 (<0.01%)	<0.001
Total ophthalmic hospitalizations	2240 (1.0%)	61 (0.8%)	29 (1.3%)	0.141

^1^ Data evaluated using a Pearson chi square test.

**Table 3 jcm-09-01271-t003:** Multivariable analyses of long-term total ophthalmic morbidity and vascular and other morbidity in offspring prenatally exposed to preeclampsia.

Variables	Adjusted HR	95% CI		*p*-Value ^1^
		Low Limit	Up Limit	
**I-Total Ophthalmic Morbidity ***				
No preeclampsia	1 (ref)			
Mild preeclampsia	0.848	0.658–1.094		0.205
Severe preeclampsia or eclampsia	1.358	0.941–1.959		0.102
Maternal age (years)	0.995	0.988–1.003		0.206
Ethnicity	0.811	0.746–0.881		<0.001
**II-Vascular and Other Ophthalmic Morbidity ****				
No preeclampsia	1 (ref)			
Mild preeclampsia	0.535	0.295–0.970		0.040
Severe preeclampsia or eclampsia	1.861	1.051–3.295		0.033
Maternal age (years)	0.992	0.979–1.005		0.230
Ethnicity	0.628	0.535–0.737		<0.001

^1^ Data evaluated by Cox proportional hazards models. * Model I -2Log Likelihood for total hospitalizations = 159,877. ** Model II -2Log Likelihood for Vascular and other = 15,926. HR, hazard ratio; CI, confidence interval.

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
