# Peer review of "Prenatal Exposure to Preeclampsia and Long-Term Ophthalmic Morbidity of the Offspring"

_jcm, 2020, doi:10.3390/jcm9051271_

Round 1

Reviewer 1 Report

I am happy with the response to my comments. I was very interested to see the difference in vascular opthalmic complications, which is a biologically plausible mechanism by which preeclampsia would impact opthalmic issues. 

I was surprised to see mild preeclampsia being "protective." I would hesitate to draw the conclusion that it is truly protective as that seems highly unlikely, and may be due to chance (4% chance of this finding purely being due to chance). So I would just suggest highlighting that mild preeclampsia was not a risk factor, but be cautious about mentioning it too much as a 'protective factor.' 

Author Response

Comments and Suggestions for Authors

I am happy with the response to my comments. I was very interested to see the difference in vascular ophthalmic complications, which is a biologically plausible mechanism by which preeclampsia would impact ophthalmic issues. 

I was surprised to see mild preeclampsia being "protective." I would hesitate to draw the conclusion that it is truly protective as that seems highly unlikely and may be due to chance (4% chance of this finding purely being due to chance). So, I would just suggest highlighting that mild preeclampsia was not a risk factor but be cautious about mentioning it too much as a 'protective factor.' 

Response:

We thank the reviewer for his insightful comment. Our findings regarding mild preeclampsia were revised to cautiously mention that in our study, mild preeclampsia was not found to be a risk factor for long-term vascular associated ophthalmic morbidity in the offspring. Additional research is required in order to further investigate these findings.

The relevant paragraph in the Discussion section was edited accordingly: "The major findings of our study indicate that in our population severe preeclampsia or eclampsia were found to be a significant risk factor for long-term vascular associated ophthalmic morbidity in offspring. Our findings are supported by several studies who have previously described the relation between the severity of preeclampsia and long-term vascular morbidities (i.e. cardiovascular, renal) in the context of the mother and the fetus. Nevertheless, in our population, mild preeclampsia was not found to be a risk factor for long-term vascular associated ophthalmic morbidity in the offspring. Additional research is required in order to further investigate these findings".

We would like to take this opportunity and express our sincere gratitude to the reviewer, who have identified areas of our manuscript that needed corrections and modifications, and we believe that these changes have significantly improved our manuscript.

Sincerely,

Prof Eyal Sheiner MD, PhD

Department of Obstetrics and Gynecology, Soroka University Medical Center, Ben-Gurion University of the Negev, Beer-Sheva

Reviewer 2 Report

This a large retrospective cohort study investigating the role of in utero exposure to preeclampsia is associated with ophthalmic disease. This is a very interesting hypothesis and very relevant, given the growing understanding of the importance of in utero life to later disease.  However, the paper suffers from a lack of clarity, particularly with respect to the methods section. Specific Comments are as follows:

General Comments:

1. It's unclear to me why the authors are interested in time-to-first-hospitalization. What does a shorter or faster time represent, clinically? Is the research question more directly assessed by simply evaluating hospitalization for ophthalmic condition (yes/no) or number of hospitalizations (though that would REALLY be evaluating severity)? If the intent is truly to investigate time-to-event, then the clinical relevance must be explained. 

Abstract:

  1. Needs to be clear that authors are investigating a time-to-event outcome.

Introduction 

  1. Discuss opposing hypothesis -- that shared underlying etiologic pathways are the reason for associations with later CVD, etc. (not changes due to PE)
  2. Lines 71-73: Rather than make a priority claim that may or may not be true (unless authors read literature in all languages), it would be better to point out the gap in the literature.  Something like, "There is a lack of published data on the ophthalmic consequences of PE exposure..."  Then you could emphasize that this study aims to fill that gap.

Experimental section

  1. Were there no HELLP syndrome cases? With whom were these combined? They may have mild or severe PE, or no hypertension at all.
  2. Lines 93-96: Curious about the numbers. How many censored due to mortality? Did it differ by exposure status? How many censored due to reaching 18? No one lost-to-follow-up?  Was there some reason you stopped at 18? Please provide rationale.
  3. Linkage of databases is not clearly explained. We are not informed that some babies were not able to be linked as they were not always seen at that hospital again (moved, etc.). There needs to be a more more thorough description of this linkage.

Statistical Analysis

  1. This section is entirely unclear and must be written with significantly more detail, such that another statistician could repeat all analyses.
  2. Lines 108-110: How are variables (e.g., maternal age) categorized? Rationale?
  3. Line 116: Not sure what authors are saying is correlated with SES. Clarify.
  4. Were maternal age and ethnicity the only covariates in the model? This is unclear in the table as well, but should be clarified in all places. This section is, overall, not at all clear such that someone could repeat it in another population (the standard for methods sections).

    Seems a bit overkill to do both Kaplan-Meier and Cox analysis, though this may be done in some studies. Generally, the choice depends partly on study design and partly on research question.  Do the authors intend to test median s(t) in the various groups (logrank test) or estimate the association between PE history and ophthalmic outcomes, adjusting for confounders? Of course, this is all moot if the authors are not truly interested in time-to-event (which must be clinically justified).

    Furthermore, it is unclear which variables were included in the model and how they were chosen for inclusion.  For example "...entered gestational age..." implies that it was a covariate, but there is no mention of such in the Cox list of adjustments.  How was confounding defined?

    Gestional Age may be a mediator between history of PE and ophthalmic complications. For example, PE was the cause of (early) GA, which then lead to the complications.  If this is true, it should not be included in any model examining PE hx and complications.  Suggest that the authors consider evaluating GA as a mediator (as a secondary aim).

    There is no mention of checking model appropriateness or fit.  No power calculation is included, but we should have been provided with the detectable effect size with the available N.
  5. Why not derive a curve of the instantaneous rate from the Cox model to use for evaluation of the assumption of proportionality? Though you mention that you will evaluate this, you have a clear violation of proportionality in Fig 2C. 

Results

  1. The differences between groups in maternal age is not clinically significant. It is only statistically so due to the large sample sizes. As stated, you give the wrong impression. It's important to distinguish these two types of significance.
  2. Results must contain effect sizes as well as p-values.  Text and tables should be mostly independent such that you could read the text OR look at the tables and get all the information required for interpretation.

  3. All tables need footnotes indicating the type of analysis reflected in the table. For example, "presented as mean plus/minus SD; p-values obtained from [stats tests]. This is important because I cannot tell which groups are being compared here -- what does the p-value reflect?

  4. Table 2, ROP: Must check this result.  I cannot see how this is significant, let at all highly so. There are virtually no clinical differences between these numbers.
  5. Line 161: And what else, if anything, was included in the model? This sentence structure implies that there are other covariates unmentioned.
  6. Table 3 "Model 2" (bad name as this implies an alternate model of the same data) -- mild and severe PE results seem contradictory and this should be further discussed as to why this may have occurred in the discussion section.

Discussion

  1. Should begin with a short summary of main findings.
  2. Lines 169- ~174 sound like introduction material.  A discussion includes (usually in this order):

    1. A comparison of your results to the relevant literature. Basically, put the results of this study into context with the published literature. (mostly complete)

    2. A proposed mechanism or biological explanation for your findings (which you have, but too early in the section as you have not yet done #1)

    3. A thorough discussion of study strengths and limitations, including the potential impact of those limitations on study results. (potential impact not discussed).

3. Lines 190-201: This paragraph should come first (minus the "in light of biological plausibilty")

4. Lines 199-200: Rather, it merely suggests that mild PE protects against the conditions the authors examined. There is no evidence provided that there are other ophthalmic conditions for which the mechanism might be different? If there are other possible ophthalmic conditions that are relevant, why weren't these included? It could easily be a spurious finding if there is no good explanation. Occam's Razor. 

5. Lines 203-204: Make this clear in methods. This will help address why no apparent loss-to-follow up. [Note: there is later mention of LTFU but this is very unclear from everything that comes before.]

6. Line 205: Would emphasize the fact that these two databases could be linked and provide long-term follow up data on babies born at that hospital.  It's a real strength.

7. Lines 206-207: Explain the potential effect on interpretation of study results, e.g., generalizability.

8. Lines 209-211: Herein lies the problem.  There is no accounting for subjects using a STROBE diagram. This must be included.  See STROBE guidelines and be sure to adhere to all reporting requirements.

9. Lines 211-212: Why? You can look at this in your data and check. 

10. Lines 214-216: Yes, a smaller N will reduce power, but this is not the concern here, as this is still a rather large number of events.  I do not think power is a concern (but no calculations are provided). I think the concern is generalizability of your results to populations who were not hospitalized.

11. Lines 218-220: Suggest delete the priority claim and opinion.  Opinions do not belong in scientific research papers.

Author Response

This manuscript is a resubmission of an earlier submission. The following is a list of the peer review reports and author responses from that submission.

Round 1

Reviewer 1 Report

The manuscript explored the association between children prenatally exposed to preeclampsia and long-term ophthalmic morbidity among offspring.

Main concerns

  • No clear definition of preeclampsia, mild preeclampsia and severe preeclampsia.
  • Preeclampsia is typically considered an increased risk among the first pregnancy. The table 1 shows that the proportions of mild preeclampsia are very similar between parity 1 and 2-4. I am wondering why?
  • What are the sensitivity and specificity of the outcomes under study in the pediatric hospitalization records? How the sensitivity and specificity of the outcomes would affect the results?
  • Uses of ANOVA tests have to fulfill the assumptions including normal distribution and the same standard deviation between groups. There is no information on whether and how the assumptions were checked?
  • For a survival analysis, it should clearly describe the starting time of the follow-up and the ending-time of the follow-up. There is no information through the manuscript.
  • The authors used the Cox regression to estimate hazard ratios. One of the critical assumptions for Cox regression is proportional hazards. Through the manuscript, there is no information on how the assumption was checked and whether the assumption was fulfilled?
  • For an observational study, a big effort is to control for potential confounders/confounding. The authors controlled only two factors including maternal age and gestational age. No information on why the two factors should be controlled and how the two factors were controlled (for example whether categorical variables or continuous variables, etc). Furthermore, preterm birth (gestational age) is a potential intermediate variable between preeclampsia and the outcomes under study. Adjusting for an intermediate variable not only may diminish the associate between the preeclampsia and the outcomes under study but also potentially introduce so-called collider stratification bias. Beside the above two factors, what about other potential confounders for example parental ophthalmic diseases, social economics etc.?
  • For a survival analysis, only providing number of cases in the table 2 is not good enough. It should additionally provide follow-up time and incidence rate.
  • Regarding overall follow-up time, the authors mentioned that it was up to 18 years. However given the study population – children born 1991 – 2014, the follow-up for for example some children born after 2010 will be much shorter. Therefore, it is important to provide the distribution of the study population based on their birth year.

Minor concerns

  • The number of severe preeclampsia or eclampsia is 2222. Please check out whether the proportion should be 0.95% or 0.92%?
  • No clear definition of the outcome under study. ICD-9 codes were used but no information on which codes were exactly selected.
  • The figure 1, the title is “Hazard function” but the legend indicates “… the cumulative incidence …”. The hazard function and cumulative incidence are not the same thing. Please be correct it and be consistent.
  • It typical say low limit- and up limit of 95%CI instead of “Min” “Max” in the table 3.
  • In the section of introduction, the authors mentioned “The short-term ophthalmic complication of preeclampsia for both mother and child have been previously described in the literature” however, no information on how the current results compared to the short-term ophthalmic complication?

Suggestions that may improve the manuscript

  • As a reader, I would have Kaplan Meier curve for each individual outcomes and combined outcome.
  • The color used by the figure 1 was not well-considered. It is not easy to read.

Reviewer 2 Report

-The study design was based on offspring who were hospitalised for opthalmic conditions. Not including community based conditions is a significant limitation. 

-It may have been better to look purely at opthalmic conditions resulting from vascular complications, as that was the main suspected mechanism (infection and inflammation would surely be expected to be similar?) 

-Why was the prevalence of preeclampsia lower than expected, only 3%? 

-The prevalence of opthalmic conditions requiring hospitalisations was so low that it could not have been expected to have had sufficient power